# Introduction and adoption of innovative invasive procedures and devices in the NHS: an in-depth analysis of written policies and qualitative interviews (the INTRODUCE study protocol)

Sian Cousins,[1] Hollie Richards,[1] Jesmond Zahra,[1] Daisy Elliott,[1] Kerry Avery,[1] Harry F Robertson,[1] Sangeetha Paramasivan,[1] Nicholas Wilson,[1] Johnny Mathews,[1] Zoe Tolkien,[1] Barry G Main,[1,2] Natalie S Blencowe,[1,2] Robert Hinchliffe,[1,3] Jane M Blazeby[1,2]

For numbered affiliations see end of article.

**Correspondence to**
Sian Cousins;
sian.cousins@bristol.ac.uk

## ABSTRACT

**Introduction** Innovation is key to improving outcomes in healthcare. Innovative pharmaceutical products undergo rigorous phased research evaluation before they are introduced into practice. The introduction of innovative invasive procedures and devices is much less rigorous and phased research, including randomised controlled trials, is not always undertaken. While the innovator (usually a surgeon) may introduce a new or modified procedure/device within the context of formal research, they may also be introduced by applying for local National Health Service (NHS) organisation approval alone. Written policies for the introduction of new procedures and/or devices often form part of this local clinical governance infrastructure; however, little is known about their content or use in practice. This study aims to systematically investigate how new invasive procedures and devices are introduced in NHS England and Wales.

**Methods and analysis** An in-depth analysis of written policies will be undertaken. This will be supplemented with interviews with key stakeholders. All acute NHS trusts in England and Health Boards in Wales will be systematically approached and asked to provide written policies for the introduction of new invasive procedures and devices. Information on the following will be captured: (1) policy scope, including when new procedures should be introduced within a formal research framework; (2) requirements for patient information provision; (3) outcome reporting and/or monitoring. Data will be extracted using a standardised form developed iteratively within the study team. Semistructured interviews with medical directors, audit and governance leads, and surgeons will explore views regarding the introduction of new invasive procedures into practice, including knowledge of and implementation of current policies.

**Ethics and dissemination** In-depth analysis of written policies does not require ethics approval. The University of Bristol Ethics Committee (56522) approved the interview component of the study. Findings from this work will be presented at appropriate conferences and will be published in peer-reviewed journals.

## Strengths and limitations of this study

► This is the first systematic study addressing how the National Health Service (NHS) introduces innovative invasive procedures into clinical practice; a topic of international public, professional and political interest given the wealth of historic and recent examples of patient harm caused by lack of regulation.

► The study combines analysis of written policies and interviews with key stakeholders to ensure an in-depth exploration of the topic to inform national guidance to improve and standardise the introduction of innovative invasive procedures and devices.

► The degree to which policies are actually adhered to across local NHS organisations is not systematically explored in the current study and is recommended for future research.

## INTRODUCTION

Invasive procedures are a fundamental part of healthcare and can include surgical operations with and without devices, as well as endoscopic and radiologically guided interventions. At least 230 million invasive procedures are delivered worldwide,[1] with 12.5 million undertaken in the UK annually.[2] This number is likely to increase with continued innovation, including the advent of new technologies and minimal access procedures.

Innovation in invasive procedures is key. It may include modifying existing techniques,[3] to performing completely new first-in-human invasive procedures.[4] While innovation is common, the governance surrounding it is not standardised and is currently under much scrutiny.[5–7] A recent inquest into the death of a patient following robot-assisted

cardiac surgery, a procedure that had not been previously performed in the UK, found insufficient governance surrounding the introduction of this innovative procedure.[6] Recommendations from the Coroner, echoed by a statement from the Royal College of Surgeons,[8] included introducing stricter governance relating to the use of new technologies and procedures, specific measures to assess the competence and training received by clinicians wishing to undertake them, and detailed patient information provision regarding the risks associated with new procedures.

Innovation in invasive procedures may be introduced under the auspices of formal research studies, with a protocol and application for ethical approval. However, multiple reviews show that ethical approval is rarely gained when delivering innovative invasive procedures.[9–12] While medical devices to be used inside the body require a European Conformity (CE) mark[13–15] prior to use in the UK, the evidence required to gain this certification does not often come from high-quality randomised controlled trials and post-marketing surveillance is minimal.[7] Although medical device regulations are improving,[16] this is very different to the tightly governed and transparent developmental pathways required for the introduction of new pharmaceuticals, including requirements for formal assessments of risk–benefit balance and post-market safety monitoring.[17]

Outside of research, innovative invasive procedures and devices may be introduced via local hospital policies. The UK National Institute for Clinical Excellence (NICE) is an independent body responsible for providing evidence-based guidance to the UK National Health Service (NHS) on health and social care. The NHS refers to the four publicly funded healthcare services in the UK (NHS in England, NHS Wales, NHS Scotland, and Health and Social care in Northern Ireland) and is made up of local NHS organisations (eg, NHS trusts in England and health boards in Wales) (online supplementary file 1). It is recommended by NICE that these local NHS organisations have appropriate governance structures in place to review, authorise and monitor the introduction of new invasive procedures.[18] In addition, NICE recommends the approval of new invasive procedures that do not have existing NICE guidance should only be given if appropriate training of those delivering the procedure is demonstrated, patients are made aware of the new status of the procedures and there are proposed arrangements for clinical audit.[19] These recommendations are echoed by organisations worldwide.[20–22] In the UK, it is the responsibility of local hospital organisations to implement this guidance, although the extent to which this occurs is unknown, and there are instances where innovative procedures are introduced into practice without any formal governance.[6] Furthermore, where local policies do exist, little is known about their content or how they are used in practice; how trusts define a new procedure (ie, when the guidance should be applied), what information should be given to patients, and how outcomes of new procedures are recorded and monitored is unclear. Examination of when policies are being applied provides valuable information about the presence or absence of local governance frameworks for the introduction of innovative invasive procedures. Furthermore, guidance related to patient information provision will provide insight into whether patients are informed about the innovative status of procedures to be delivered. It is also important that outcomes are routinely and effectively monitored to support their continued use or to ensure those that are ineffective and/or unsafe are abandoned.

A systematic literature review conducted by the Australian Safety and Efficacy Register of New Interventional Procedures—Surgical[23] identified only six publications related to how acute healthcare organisations introduce new invasive procedures. These included retrospective case reports of new procedures being introduced in individual hospitals in the UK[24] and Australia,[25] and case studies of qualitative interviews with surgeons and clinicians at hospitals in Canada regarding how decisions to introduce new technologies were made.[26 27] Findings indicated that the introduction of new procedures and technologies were based predominantly on surgeons' perceptions that such innovations would improve patient outcomes, safety and care, with no structured decision-making process in place at an institutional/governance level. To date, there has been no comprehensive review of current local NHS policies, many of which are inaccessible by traditional literature systematic searches used in the above review.

### Aim

This study aims to undertake an in-depth analysis of local NHS policies to establish the governance in place for the introduction of new invasive procedures and devices. This will include examination of (1) how policies outline scope for their use, including how they define which invasive procedures and devices are eligible under their remit (eg, new or modified) and guidance given about when research approvals should be sought; (2) recommendations for patient information provision; and (3) processes for monitoring and reporting outcomes of innovative procedures and devices, including how decision-making regarding adoption or stopping of the procedure or device are made.

### METHODS AND ANALYSIS

This work will comprise two parts occurring concurrently and iteratively: (1) systematic analysis of written local NHS policies for the introduction of new invasive procedures and devices; (2) interviews with key stakeholders (eg, medical directors, audit and governance leads, and surgeons) regarding surgical innovation in practice, including knowledge of and implementation of current policies.

### Systematic analysis of NHS policies for the introduction of new invasive procedures and devices

#### Sampling and data collection

All acute NHS trusts in England (n=150) and NHS Health Boards in Wales (n=7)[27] will be systematically approached

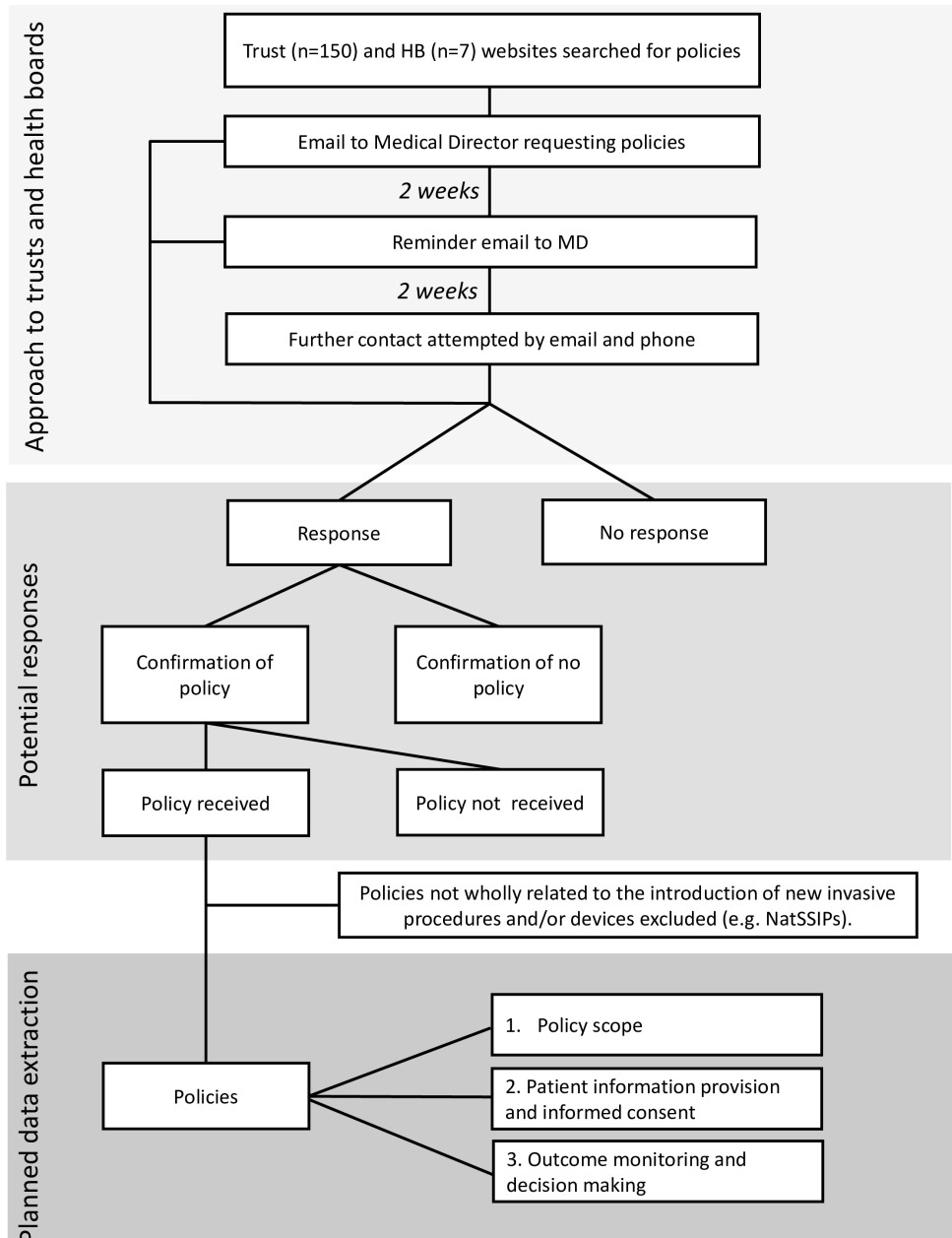

**Figure 1** Approach to trusts and health boards and planned data extraction from policies for the introduction of new invasive procedures and devices. HB, Health Board; NatSSIPs, National Safety Standardsfor Invasive Procedures.

(figure 1). Initially, online searches will be performed to determine if policies for the introduction of new invasive procedures and/or devices into clinical practice are available online. An online search engine (Google) will be used to locate the website for each NHS trust/health board (HB). The individual trust/HB website search function will be used to determine if the organisation has a copy of the relevant policy available online using search terms such as 'new procedure' and 'policy/policies'. Exploratory searching of the trust/HB website will also be conducted to ensure relevant information is not missed. Where policies are not available, local NHS organisations will be approached as follows. Medical Directors will be emailed by senior authors (JMB or RH) to request copies of written policies. Non-responders to the initial

email will have one email reminder 2 weeks later and will then be approached by email and phone by a senior research associate and a research fellow. Trusts/HBs not responding to this will have one final email from the original project lead. Policies not wholly related to the introduction of new invasive procedures and/or devices will be excluded (eg, device management policies, National or Local Safety Standards for Invasive Procedures (NatSSIPs, LocSSIPs)). Invasive procedures will be defined as procedures where access is gained via an incision, natural orifice or percutaneous puncture or involving devices used inside the body.

Trust/HB demographics, including geographical area and acute trust type (England only, eg, small, specialist, teaching, foundation status) will be collected.[28]

### Data extraction from trust policies

A data extraction form will be developed using a priori themes drawn from the literature and knowledge of the area among team members (see online supplementary file 2 for a preliminary data extraction form), and themes that inductively emerge from the initial coding of a subset of documents independently by two researchers.

The form will be piloted on a sample of policies gathered during scoping work until the team is satisfied that it fully captures all initial key themes. The finalised form will be converted into an electronic database[29] where data will be imputed directly from written policies. All policy documents, including addendums, will be systematically examined. Ten per cent of policies will be randomly selected for double independent data extraction to maximise reliability of data extraction. Where there are discrepancies, these will be resolved through consensus, and where this is not achieved, a third independent reviewer will be consulted.

Data will be extracted about (1) policy scope, (2) patient information provision and consent, and (3) outcome monitoring and decision-making (detailed below). For each policy, we will also extract details of policy title and date, date of last policy review and date of next policy review. In addition, if a formal committee is named in the policy as being involved in reviewing the introduction of a new procedure, we will collect data on the title of the committee, committee chair, core members and frequency of meetings.

#### Policy scope

How policies define which invasive procedures and devices are eligible under their remit (eg, policy definitions of new procedures) and guidance regarding how to decide whether the new invasive procedure/device requires 'research approvals', in addition to or instead of local policy approvals alone, will be captured. Policy scope data will not be used to inform inclusion or exclusion of policies from analysis.

#### Patient information provision and consent

Guidance about specific consent procedures and how patients should be informed when a procedure is new, modified or being conducted by a clinician/in a trust for the first time will be captured. Specific requirements relating to patient information leaflets (PILs), the submission of these PILs to the trust for evaluation and any processes in place to monitor adherence to guidance regarding patient information will also be extracted.

#### Outcome monitoring and decision-making

In addition to outcomes typically associated with clinical effectiveness studies, outcomes of specific relevance to evaluating innovation of invasive procedures and devices will be extracted; these may include operator experiences, intended function (eg, benefit) of the new/modified procedure/device, unanticipated harms, and failure and/or abandonment of the procedure/device. Mechanisms proposed for outcome monitoring (eg, registered audit, feedback to the committee) will be captured, in addition to guidance for decision-making regarding when procedures should be introduced into routine clinical practice or abandoned, and recommendations for wider reporting.

### Data analyses (systematic analysis of NHS trust policies)

In order to gain an overview of the scope of the policies and develop an in-depth understanding of specific themes of relevance to the study aims, mixed-methods analyses will be undertaken.

Trust/HB demographics including geographical area and acute trust type will be tabulated. Quantitative data including response rates, and the presence or absence of a new invasive procedures/devices policy within each local NHS organisation will be presented.

Data related to each of the three key data extraction areas outlined above will be tabulated and descriptive statistics provided. For example, the number of policies that provide guidance for when procedures should only be conducted within a research study will be counted.

Free text data related to each of the three data extraction areas will be extracted from policies and analysed as following. Verbatim sections of policies will be transferred to qualitative data analysis software (NVivo, V.11) and analysed thematically in several key stages: (1) two researchers will independently code a subset of policy documents to develop a preliminary coding frame; (2) any discrepancies will be resolved through discussion before the coding frame is applied to the full dataset; (3) codes will be grouped into themes and subthemes by examining commonalities, differences and relationships in the data; (4) themes will be regularly reviewed to ensure they accurately encapsulate the data. Findings from this qualitative analysis will be written up descriptively.

### Interviews with key stakeholders

In-depth semistructured interviews will be conducted with representatives from key stakeholder groups as detailed below.

### Recruitment and sampling

Professionals involved in local governance processes related to the introduction of new procedures and devices (eg, new procedures committee members, medical directors) and healthcare professionals with experience of introducing new/modified procedures or devices into clinical practice (eg, surgeons, nurses) will be identified from policy documents, trust websites and clinical contacts of the research team. A snowball sampling approach,[30] where interviewees are asked to recommend the names of other potential interviewees, will be used to facilitate recruitment. To ensure maximum variation within the sample,[30] we will interview participants from varying geographical locations, trust types, different

surgical specialities and from trusts with and without policies. Participant characteristics will be reviewed as recruitment and analyses are ongoing throughout the study, and under-represented groups or individuals with particular knowledge and/or experiences of particular interest will be purposively sampled.[30] It is anticipated that up to 60 stakeholders will be interviewed, although data analysis will be driven by the objective of achieving data saturation (where no new themes emerge from the data).

### Data collection

Written informed consent will be obtained from all participants before interviews commence. Interviews will be conducted by one of two experienced qualitative researchers, either via telephone or in person, at times convenient for participants. Interviews will be audio-recorded using encrypted audio-recording devices (Olympus DS3500). Discussions will follow a topic guide that will vary by stakeholder group so that key issues are covered, while ensuring participants are able to talk about new issues they feel are important. Topic guides (see online supplementary file 3 for example preliminary topic guide for clinicians) will be adapted iteratively as analyses of interviews and written policies progresses so that any emerging issues can be discussed with subsequent participants.

Interviews with professionals involved in governance processes will specifically explore how new procedures are introduced. This will include how policies define the types of technologies and procedures that will be reviewed for introduction to the trust (with examples if possible); the approval processes in place; and methods for follow-up, monitoring of approved technologies/procedures and abandonment of procedures. Interviews with healthcare professionals will explore their views on surgical innovation, monitoring of new procedures/devices and, when applicable, their experience of introducing a new procedure or device into clinical practice.

### Analysis (interviews with key stakeholders)

Interviews will be transcribed in full and verbatim, checked against the original recording for accuracy, and imported into NVivo (V.11). Data will then be systematically assigned codes and analysed thematically using constant comparative techniques.[31] A subset of the transcripts will be doubled coded by a second qualitative researcher, with any discrepancies in coding discussed and resolved. Data collection and analysis will proceed in parallel, with emerging findings informing further sampling (theoretical sampling) and data collection. Dissonant views that challenge the emerging dominant perspectives will be actively pursued (negative case analysis) to ensure the inclusion of diverse viewpoints. Descriptive accounts of the data, which take into consideration of the views and background of the analysts, will then be written.

### Data protection and confidentiality

All data relating to participants' personal identities will be anonymised using unique study identifiers. This data will be stored in a separate encrypted file, in a separate location from the study data on the University of Bristol server. It will only be accessible to the research team and used only in the event of re-contacting study participants to verify information, for example, quotes. Verbatim quotations that may be used for publications or presentations will also be anonymised.

## ETHICS AND DISSEMINATION

The in-depth analysis of written policies does not require ethics approval, in accordance with the Health Research Authority definition of research.[32] The qualitative component of the work has been approved by the University of Bristol Ethics Committee (reference 56522).

### Expected outcomes of the study

This work will provide an in-depth exploration and summary of current governance procedures for clinicians wishing to introduce innovative invasive procedures and devices into NHS practice in England and Wales. Understanding how innovative invasive procedures are introduced will identify limitations in current guidance, inform the development of standardised guidance and raise hypotheses for future research. It is expected that this work will inform national guidelines regarding the introduction of innovative invasive procedures.

### Dissemination

Findings from this work will be presented at appropriate conferences and published across several papers highlighting main findings. These will include in-depth analyses of the scope of written policies, including when innovative procedures should be delivered within a research governance framework, and how policies define 'new' and 'modified' procedures/devices. Additional publications will focus on guidance related to patient information, consent and outcome monitoring after the introduction of new procedures and devices. Additionally, qualitative data from interviews with key stakeholders regarding surgical innovation in practice, including knowledge and implementation of current policies, will be published separately.

## DISCUSSION

Currently, little is known about the ways in which new invasive procedures and devices are introduced into NHS clinical practice outside the context of research. This is a topical issue as several concerning and problematic high-profile cases have emerged recently.[5–7 33] This study will systematically study current NHS practice to determine the presence, content and implementation of policies relating to the introduction of new invasive procedures and devices in NHS England and Wales.

Identified policies will be scrutinised to determine what guidance is given regarding policy scope, patient information provision and the monitoring, reporting and review of outcomes. Additionally, interviews with stakeholders (such as surgeons and members of new procedures committees) will further inform this work.

The ways in which new procedures, including surgery, are introduced into practice is a topic of international public interest. This study is the first of its kind and it is anticipated it will inform future NHS governance and practice in this field.

## PATIENT AND PUBLIC INVOLVEMENT

The current study comprises a core component of the work undertaken within the National Institute for Health Research (NIHR) Bristol Biomedical Research Centre (BRC) Surgical Innovation theme, which aims to improve the safe and transparent translation of innovative procedures/devices to clinical practice. A patient and public involvement (PPI) group has been established as part of the NIHR Bristol BRC, where patients who have undergone surgery are asked about their views regarding how new surgical procedures are undertaken in NHS clinical practice. This involves discussion around what information patients would like to be provided with before and after receiving a new invasive procedure/device, and what health/lifestyle outcomes after surgery would be considered important. To date, the consensus between PPI group members is that the work being undertaken to improve the way in which new procedures are introduced into clinical practice is important and could have positive implications for future healthcare in the NHS. The current study comprises a first step in the process of improving how new procedures are introduced into practice and the PPI group will continue to provide input throughout the study and future work to develop related guidance and disseminate findings.

**Author affiliations**
[1]National Institute for Health Research Bristol Biomedical Research Centre Surgical Innovation Theme and the Medical Research Council ConDuCT-II Hub for Trials Methodology Research, Bristol Centre for Surgical Research, Bristol Medical School: Population Health Sciences, University of Bristol, Bristol, UK
[2]Division of Surgery, University Hospitals Bristol NHS Foundation Trust, Bristol, UK
[3]Vascular Services, North Bristol NHS Trust, Westbury on Trym, UK

**Authors' contributions** All authors contributed to the development of the idea and drafting and revision of the manuscript. JMB is the lead of the NIHR Bristol Biomedical Research Centre Surgical Innovation theme and formed the methodological ideas to understand surgical innovation with contributions from SC, HR, JZ, DE, KA, HFR, SP, NW, JM, ZT, BGM, NSB and RH. All authors gave approval for the manuscript to be submitted.

**Funding** This study was supported by the NIHR Biomedical Research Centre at University Hospitals Bristol NHS Foundation Trust and the University of Bristol, the MRC ConDuCT-II (Collaboration and innovation for Difficult and Complex randomised controlled Trials In Invasive procedures) Hub for Trials Methodology Research (MR/K025643/1) (www.bristol.ac.uk/population-health-sciences/centres/conduct2) and a NIHR senior investigator award (NF-SI-0514-10114).

**Disclaimer** The views expressed in this publication are those of the authors and not necessarily those of the NHS, the National Institute for Health Research, the Department of Health and Social Care or the MRC.

**Competing interests** None declared.

**Patient consent for publication** Not required.

**Provenance and peer review** Not commissioned; externally peer reviewed.

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
