## [Reviewer comments · BMJ Open]

ARTICLE DETAILS

TITLE (PROVISIONAL)	The introduction and adoption of innovative invasive procedures and devices in the NHS: an in-depth analysis of written policies and qualitative interviews (the INTRODUCE study protocol)
AUTHORS	Cousins, Sian; Richards, Hollie; Zahra, Jesmond; Elliott, Daisy; Avery, Kerry; Robertson, Harry; Paramasivan, Sangeetha; Wilson, Nicholas; Mathews, Johnny; Tolkien, Zoe; Main, Barry; Blencowe, Natalie; Hinchliffe, Robert; Blazeby, Jane

VERSION 1 – REVIEW

REVIEWER	Akihide Konishi PMDA, Japan
REVIEW RETURNED	04-Mar-2019

GENERAL COMMENTS	I can't understand the purpose of this study. Please explain how to collect and evaluate the data with figure and please explain the justification of the data to evaluate. Furthermore, how these analyzed data are going to be used (e.g. partial change for application of the medical devices) or what will the analyzed data contribute? P4, L26; CE mark does not necessary require high quality RCT and PMS for new pharmaceutical procedure which is different from US and Japan. Please mention about fundamental policy of evaluating risk benefit balance of new pharmaceutical procedure and post-market risk management measures. I heard CE mark has recently changed to strick assessment at approval process than before because of several cases with serious adverse event at post marketing stage. Please mention background about this in this paper. Please illustrate the relationship among CE, NICE and NHS in England simply with figure. It is difficult for reader living in other country to understand. Further, please illustrate the task and the role of NHS with figure simply. Abbreviation must be spelled out at their initial appearance, followed by the abbreviation in parentheses. For example, NHS, JMB, RH, MRC, etc
---

REVIEWER	Dr Van Bruwaene Siska AZ Groeninge, Kortrijk, Belgium
REVIEW RETURNED	22-Mar-2019

GENERAL COMMENTS	Study addresses a very important gap in surgical literature. Goals and strategy are clearly defined. Looking forward to the results.
--

REVIEWER	Keith Isaacson Newton Wellesley Hospital, Harvard Medical School, USA
REVIEW RETURNED	20-Apr-2019
GENERAL COMMENTS	This is the first half of a manuscript that contains no results. It provides an excellent description of the background, the specific aims and the methods but no study results since the study is yet to be performed,

VERSION 1 – AUTHOR RESPONSE

Reviewer: 1

Reviewer Name: Akihide Konishi

Institution and Country: PMDA, Japan

1. I can't understand the purpose of this study. Please explain how to collect and evaluate the data with figure

Reply: We are sorry that the purpose of the study was unclear. We have altered the text in the introduction and the aim to clarify this. Also, please find now included Figure 1, which outlines in detail data collection, including approach strategy, potential responses from trusts and health boards and key data extracted from written policies.

Revision: We now include an additional Figure 1. Text in the introduction and aim has also been updated to clarify the purpose of the study -

“While medical devices to be used inside the body require a European Conformity (CE) mark¹³⁻¹⁵ prior to use in the UK, the evidence required to gain this certification does not often come from high-quality randomised controlled trials and post-marketing surveillance is minimal.⁷ Although medical device regulations are improving,¹⁶ this is very different to the tightly governed and transparent developmental pathways required for the introduction of new pharmaceuticals, including requirements for formal assessments of risk-benefit balance and post-market safety monitoring.¹⁷

Outside of research, innovative invasive procedures and devices may be introduced via local hospital policies. The UK National Institute for Clinical Excellence (NICE) is an independent body responsible for providing evidence-based guidance to the UK National Health Service (NHS) on health and social care. The NHS refers to the four publicly funded healthcare services in the UK (NHS in England, NHS Wales, NHS Scotland and Health and Social care in Northern Ireland) and is made up of local NHS organisations (e.g. NHS trusts in England and health boards in Wales) (supplementary file 1). It is recommended by NICE that these local NHS organisations have appropriate governance structures in place to review, authorise and monitor the introduction of new invasive procedures.¹⁸”

“Aim: To undertake an in-depth analysis of local NHS policies to establish the governance in place for the introduction of new invasive procedures and devices. This will include examination of: I) how policies outline scope for their use, including how they define which invasive procedures and devices are eligible under their remit (e.g. new or modified) and guidance given about when research approvals should be sought, II) recommendations for patient information provision, III) processes for monitoring and reporting outcomes of innovative procedures and devices, including how decision-making regarding adoption or stopping of the procedure or device are made.”

2. and please explain the justification of the data to evaluate.

Reply: Please find now included on page 5, paragraph 1 explicit justification for the investigation of each of the 3 key data extraction areas (1. Policy scope; 2. Patient information provision; 3. Outcome monitoring).

Revision:

“Examination of when policies are being applied provides valuable information about the presence or absence of local governance frameworks for the introduction of innovative invasive procedures. Furthermore, guidance related to patient information provision will provide insight into whether patients are informed about the innovative status of procedures to be delivered. It is also important that outcomes are routinely and effectively monitored to support their continued use or to ensure those that are ineffective and/or unsafe are abandoned.”

3. Furthermore, how these analyzed data are going to be used (e.g. partial change for application of the medical devices) or what will the analyzed data contribute?

Reply: Thank you for this comment. The text has now been amended to present more clearly how the study data will be used and how it will contribute to this area of research and clinical practice.

Revision: We have altered the text on page 12, paragraph 2, which now reads –

“This work will provide an in-depth exploration and summary of current governance procedures for clinicians wishing to introduce innovative invasive procedures and devices into NHS practice in England and Wales. Understanding how innovative invasive procedures are introduced will identify limitations in current guidance, inform the development of standardised guidance and raise hypotheses for future research. It is expected that this work will inform national guidelines regarding the introduction of innovative invasive procedures.”

4. P4, L26; CE mark does not necessary require high quality RCT and PMS for new pharmaceutical procedure which is different from US and Japan. Please mention about fundamental policy of evaluating risk benefit balance of new pharmaceutical procedure and post-market risk management measures. I heard CE mark has recently changed to strick assessment at approval process than before because of several cases with serious adverse event at post marketing stage. Please mention background about this in this paper.

Reply: Thank you for highlighting this issue. We have now outlined this background on page 4, paragraph 2. We now explicitly include mention of the policy of evaluating risk benefit balance and post-market risk management measures when introducing new pharmaceuticals. We agree that CE marking for devices does not necessarily require high quality RCT and post-market surveillance and we have now stated this in the text. The recent improvements to European Union regulations for medical devices are also now referenced – thank for your bringing this to our attention.

Revision: Text has been altered in the introduction and now reads -

“While medical devices to be used inside the body require a European Conformity (CE) mark¹³⁻¹⁵ prior to use in the UK, the evidence required to gain this certification does not often come from high-quality randomised controlled trials and post-marketing surveillance is minimal.⁷ Although medical device regulations are improving,¹⁶ this is very different to the tightly governed and transparent developmental pathways required for the introduction of new pharmaceuticals, including requirements for formal assessments of risk-benefit balance and post-market safety monitoring.¹⁷”

5. Please illustrate the relationship among CE, NICE and NHS in England simply with figure. It is difficult for reader living in other country to understand. Further, please illustrate the task and the role of NHS with figure simply.

Reply: Thank you for this helpful comment. We have now included additional text on page 4, paragraph 3 clarifying the role and function of NICE and the NHS in the UK and their relationship. We also include a simple figure as supplementary material. The CE certification is a requirement for all devices to be used inside the body in clinical practice in the UK but does not directly form part of the relationship between NICE and the NHS. This is clarified in the updated text.

Revision: Text in the introduction now reads -

“Outside of research, innovative invasive procedures and devices may be introduced via local hospital policies. The UK National Institute for Clinical Excellence (NICE) is an independent body responsible for providing evidence-based guidance to the UK National Health Service (NHS) on health and social care. The NHS refers to the four publicly funded healthcare services in the UK (NHS in England, NHS

Wales, NHS Scotland and Health and Social care in Northern Ireland) and is made up of local NHS organisations (e.g. NHS trusts in England and health boards in Wales) (supplementary file 1). It is recommended by NICE that these local NHS organisations have appropriate governance structures in place to review, authorise and monitor the introduction of new invasive procedures.17”

6. Abbreviation must be spelled out at their initial appearance, followed by the abbreviation in parentheses. For example, NHS, JMB, RH, MRC, etc

Reply: Apologies for this oversight. We have now spelled out all abbreviations.

Reviewer: 2

Reviewer Name: Dr Van Bruwaene Siska

Institution and Country: AZ Groeninge, Kortrijk, Belgium Please state any competing interests or state 'None declared': None declared.

1. Study addresses a very important gap in surgical literature. Goals and strategy are clearly defined. Looking forward to the results.

Reply: Thank you for this supportive comment

Reviewer: 3

Reviewer Name: Keith Isaacson

Institution and Country: Newton Wellesley Hospital, Harvard Medical School, USA Please state any competing interests or state 'None declared': None Declared

1. This is the first half of a manuscript that contains no results. It provides an excellent description of the background, the specific aims and the methods but no study results since the study is yet to be performed

Reply: We have clarified with the journal that it does publish protocols.